# Ni-catalyzed regio- and stereo-defined intermolecular cross-electrophile dialkylation of alkynes without directing group

Yi-Zhou Zhan[1], Nan Xiao[1] & Wei Shu [1✉]

The development of straightforward synthesis of regio- and stereodefined alkenes with multiple aliphatic substituents under mild conditions is an unmet challenge owing to competitive *β*-hydride elimination and selectivity issues. Herein, we report the nickel-catalyzed intermolecular cross-dialkylation of alkynes devoid of directing or activating groups to afford multiple aliphatic substituted alkenes in a *syn*-selective fashion at room temperature. The combination of two-electron oxidative cyclometallation and single-electron cross-electrophile coupling of nickel enables the *syn*-cross-dialkylation of alkynes at room temperature. This reductive protocol enables the sequential installation of two different alkyl substituents onto alkynes in a regio- and stereo-selective manner, circumventing the tedious preformation of sensitive organometallic reagents. The synthetic utility of this protocol is demonstrated by efficient synthesis of multi-substituted unfunctionalized alkenes and diverse transformations of the product.

[1] Shenzhen Grubbs Institute, Department of Chemistry, and Guangdong Provincial Key Laboratory of Catalytic Chemistry, Southern University of Science and Technology, 518055 Shenzhen, Guangdong, P. R. China. ✉email: shuw@sustech.edu.cn

Multiple substituted alkenes are pivotal subunits which are commonly occurring in natural products[1–3], bioactive molecules[4–7], and material sciences[8], and serve as useful precursors for other functional groups and chiral centers[9–12]. In addition, alkenes exhibit orthogonal reactivity with respect to polar functional groups, providing an opportunity for the late-stage derivatization of complex molecules. To this end, synthesis of olefins with stereo-selectivity control has become one of the major themes in the arena of synthetic chemistry and has evolved from Wittig reaction and elimination reactions to carbometallation-coupling reactions[13–15] and olefin metathesis[16–18]. Despite numerous progress in addressing this issue, stereodefined synthesis of tri- and tetrasubstituted alkenes remains a formidable synthetic challenge. On the other hand, Ni-catalyzed carbo-difunctionalization of alkynes represents an ideal route to access multiple substituted alkenes with molecular complexity via the spontaneous formation of two vicinal C–C bonds. Among which, alkylation of alkynes is more challenging due to the difficulties in forging $C_{sp2}$-$C_{sp3}$ bond[19–29]. In general, three distinct strategies have been employed to address this issue. The first strategy is a Ni-catalyzed radical addition onto alkynes, followed by recombination and coupling with arylmetallic reagents, favoring *anti*-selectivity which is dominated by steric repulsion of vinyl radicals with nickel center (Fig. 1a, top)[19–21]. The use of terminal alkynes offers limited opportunity to afford trisubstituted alkenes. The second strategy is a carbometallation of alkynes to generate vinyl nickel species, which is followed by an electrophilic quench, favoring a *syn*-selectivity (Fig. 1a, bottom)[22–31]. This method suffers from regioselectivity issues, and could be only applied to electronically and sterically biased alkynes. In most cases, directing groups are required to improve regioselectivity. These two strategies require sensitive organometallic reagents, and can be applied to alkylarylation and diarylation of alkynes, dialkylation is inaccessible due to the difficulties in forming the second $C_{sp2}$-$C_{sp3}$ bond[28]. The other is oxidative cyclization of Ni (0) complex with alkyne and a second π-component to form Ni(II) cyclic intermediates, followed by σ-bond metathesis or transmetalation with organometallic reagents (Fig. 1b) to forge the second carbon-carbon bond. Montgomery[32–40], Jamison[41,42], Ikeda and Sato[43], Ogoshi[44,45], as well as others[46–48] extensively investigated the tandem oxidative cyclization of alkynes with C = C/C = N/C = O and coupling using different organometallic reagents (R-M, M = Zn, B, Mg, Zr, Sn) to give diverse substituted alkenes (Fig. 1b, top). In 2018, Montgomery reported a seminal reductive Ni-catalyzed intramolecular oxidative cyclization of internal alkyne with

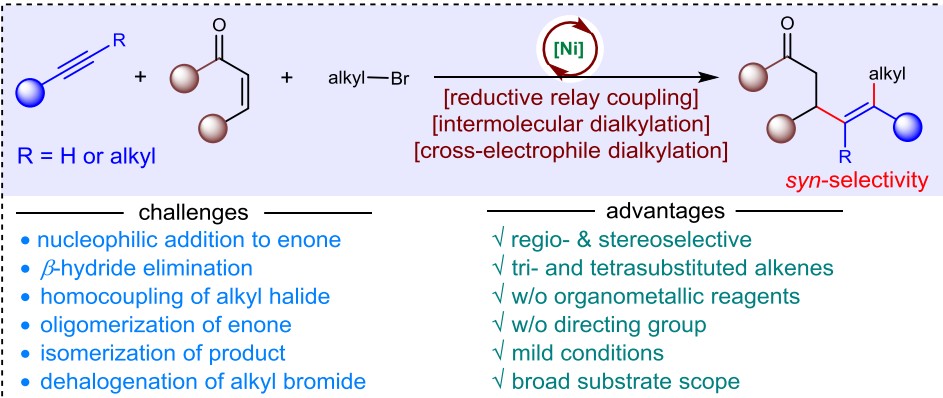

**a** Ni-catalyzed intermolecular carbo-difunctionalization of alkyne

**b** Ni-catalyzed cross-alkyl-functionalization of alkyne via cyclometallation

**c** Ni-catalyzed stereodefined intermolecular dialkylation of alkyne (This work):

**Fig. 1 Stereodefined access to multiple alkylated alkenes. a** Ni-catalyzed intermolecular carbo-difunctionalization of alkyne. **b** Ni-catalyzed cross-alkyl-functionalization of alkyne via cyclometalation. **c** Ni-catalyzed stereodefined intermolecular dialkylation of alkyne.

aldehyde, followed by coupling with alkyl bromide to afford tetrasubstituted allylic ether with cyclic substituent (Fig. 1b, bottom)[49]. The reaction allows for the use alkyl halides as alkylating reagents instead of organometallic reagents. To the best of our knowledge, no general method for fully inter-molecular cross-dialkylation of alkynes is developed, probably due to non-productive β-H hydride elimination during coupling process and competitive selectivity issues. In this context, we envisioned a fully intermolecular oxidative cyclometallation of alkyne with alkene[50–56], followed by cross-electrophile coupling of alkyl halide to sequentially forge two $C_{sp2}$-$C_{sp3}$ bonds to deliver general tri- and tetrasubstituted alkenes (Fig. 1c). This conceptual strategy arises several significant challenges: the nucleophilic addition of organonickel/zinc species onto enones, β-H elimination of alkylnickel intermediate, homocoupling/ dehalogenation of alkyl halides and oligomerization of enone under reductive conditions, and isomerization of the desired product to conjugated enone. In this work, we demonstrate the Ni-catalyzed intermolecular cross-dialkylation of alkynes with-out organometallic reagents. This mild method allows for the regio- and stereo-selective conversion of both terminal and internal alkynes into tri- and tetrasubstituted alkenes with up to four aliphatic substituents in the absence of any directing group.

## Results

**Reaction optimization**. With these concerns in mind, we set out to examine the hypothesis using 1-hexyne (**1a**), 2-cyclohexen-1-one (**2a**) and 1-bromobutane (**3a**) as prototype substrates (Table 1). After extensive evaluation of a series of reaction parameters, we identified the use of NiBr₂·DME (10 mol%), 4-cyanopyridine (5 mol%, **A1**), cerium chloride (0.8 equiv), potassium iodide (2.0 equiv), zinc (4.5 equiv) in the presence of water (1.5 equiv) in N,N-dimethylformamide (0.1 M) at room temperature as the optimal conditions, affording the desired three-component sequential dialkylation product **4a** in 66% yield (Table 1, entry 1). The reaction demonstrated exclusive regio- and syn-selectivity, which is complementary to the regio- and stereo-selectivity of Ni-catalyzed radical addition pathway. Similar result was obtained when presynthesized precatalyst was used, giving the desired product **4a** in 64% yield. The reaction showed lower efficiency in the absence of **A1**, affording **4a** in 37% yield (Table 1, entry 2). Increasing the loading of **A1** to 10 mol% and 20 mol% did not further improve the yield of **4a** (Table 1, entries 3 and 4). Replacing **A1** with pentafluoropyridine (**A2**) gave trialkylated alkene **4a** in 34% yield (Table 1, entry 5). The use of other nickel catalyst precursors, such as NiBr₂, NiCl₂·DME, and Ni(COD)₂, led to inferior results (Table 1, entries 6–8). No desired product was detected in the absence of nickel catalyst or sacrificial reductant zinc (Table 1, entry 9). The presence of cerium chloride and potassium iodide are both crucial for the three-component reaction, trace amount of desired product was formed without either additive (Table 1, entry 10). The role of KI is likely to undergo halide exchange to generate alkyl iodide in situ slowly from alkyl bromide to maintain alkyl iodide in low concentration during the reaction course. Other Lewis acids proved to be unsuccessful in this reaction (Table 1, entry 11). Water plays a critical role as a mediator in this reaction, giving the optimal result with 1.5 equiv of water (Table 1, entries 12 and 13). Evaluation of solvent effect revealed that polar solvent is required for this reaction. The reaction delivered **4a** in 23% and 43% yields, respectively, when the reaction was conducted in NMP or DMA (Table 1, entries 14 and 15).

**Substrate scope**. With the optimized conditions in hand, we turned to examine the scope of this transformation. To our delight, this reaction protocol is applicable to a wide variety of functionalized alkynes, alkyl bromides, and enones, furnishing various advanced tri- and tetrasubstituted alkenes with multiple

**Table 1 Condition evaluation for the fully intermolecular cross-electrophile dialkylation of alkynes.**

| Entry | Variation from "standard conditions"[a] | Yield of 4a[b](%) |
|---|---|---|
| 1 | None | 66 (64), 64[c] |
| 2 | w/o **A1** | 37 |
| 3 | **A1** (10 mol%) | 62 |
| 4 | **A1** (20 mol%) | 59 |
| 5 | **A2** instead of **A1** | 34 |
| 6 | NiCl₂·DME instead of NiBr₂·DME | 55 |
| 7 | NiBr₂ instead of NiBr₂·DME | 45 |
| 8 | Ni(COD)₂ instead of NiBr₂·DME | 36 |
| 9 | w/o "Ni" or Zn | NR |
| 10 | w/o CeCl₃ or KI | Trace |
| 11 | ZnBr₂/ZnCl₂/TMSCl instead of CeCl₃ | Trace |
| 12 | w/o H₂O | 17 |
| 13 | H₂O (2 equiv) | 51 |
| 14 | NMP instead of DMF | 23 |
| 15 | DMA instead of DMF | 43 |

[a]Reaction was run using 0.4 mmol of **1a**, 0.2 mmol of **2a**, and 0.4 mmol of **3a** under indicated conditions for 48 h.
[b]Yield was determined by GC analysis using n-dodecane as internal standard. Isolated yield is shown in the parentheses. **A1** = 4-cyanopyridine, **A2** = pentafluoropyridine.
[c]NiBr₂·DME and **A1** were presynthesized as the catalyst.

**Scope for alkynes**

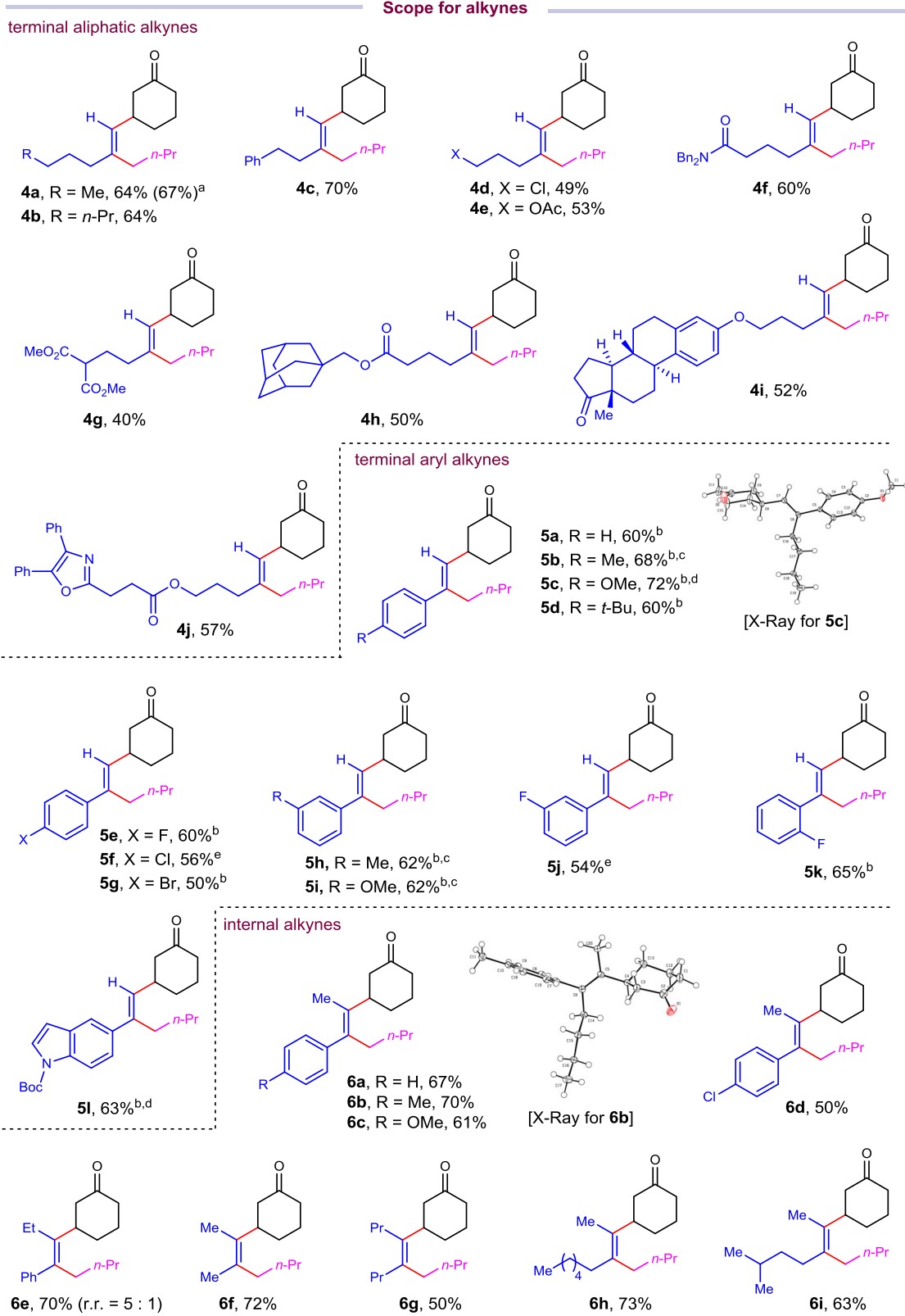

**Fig. 2 Scope for the *syn*-cross-electrophile dialkylation of alkynes with respect to alkynes.** Unless otherwise noted, isolated yield is shown on 0.2 mmol scale under standard conditions. See Table 1, entry 1 for detail. [a]Isolated yield on 10.0 mmol scale. [b]NiBr$_2$·DME (10 mol%), **A2** (20 mol%), Zn (2.5 equiv), KI (2.0 equiv), ZnBr$_2$ (10 mol%), 2-methyl-2-butanol (2.0 equiv), DMF (0.1 M), rt, overnight. [c]Alkyne was added in two portions. 1.0 equiv alkyne was added after 5 h. [d]Alkyne was added in two portions. 1.0 equiv alkyne was added after 3 h. [e]Alkyne (2.0 equiv), enone (0.2 mmol), alkyl bromide (3.5 equiv), NiBr$_2$·DME (10 mol%), Zn (5.0 equiv), CsI (5.0 equiv), ZnI$_2$ (30 mol%), 2-methyl-2-butanol (3.0 equiv), DMF (0.05 M) and 1.0 equiv alkyne was added after 3 h, rt, overnight.

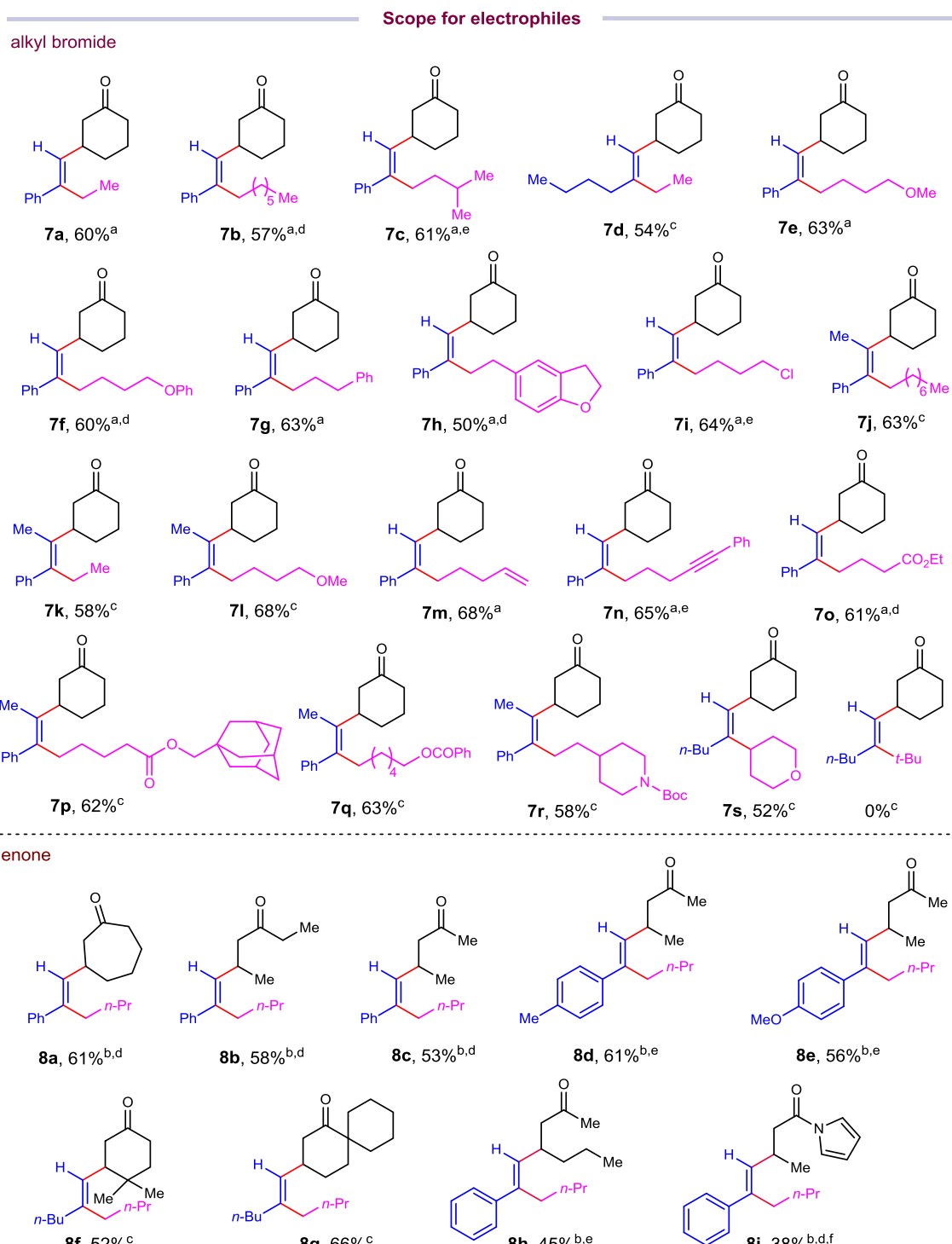

**Fig. 3 Scope for the *syn*-cross-electrophile dialkylation of alkynes with respect to alkyl electrophiles.** Unless otherwise noted, isolated yield is shown on 0.2 mmol scale under standard conditions. See Table 1, entry 1 for detail. [a]NiBr$_2$·DME (10 mol%), **A2** (20 mol%), Zn (2.5 equiv), KI (2.0 equiv), ZnBr$_2$ (10 mol%), 2-methyl-2-butanol (2.0 equiv), DMF (0.1 M), rt, overnight. [b]The reaction was conducted using 0.2 mmol of enone (1.0 equiv), alkyne (2.8 equiv), *n*-BuBr (3.5 equiv) in the presence of NiBr$_2$·DME (10 mol%), Zn (3.0 equiv), CsI (5.0 equiv), 2-methyl-2-butanol (2.0 equiv), and ZnI$_2$ (30 mol%) in DMF (0.05 M). [c]Standard conditions were used. [d]Alkyne was added in two potions. 1.0 equiv of alkyne was added after 5 h. [e]Alkyne was added in two portions. 1.0 equiv of alkyne was added after 3 h. [f]Yield based on the recovery of enone.

aliphatic substituents in excellent levels of regio-, chemo-, and stereo-selectivity (Figs. 2 and 3). First, the scope of alkynes was investigated under the reaction conditions (Fig. 2). Aliphatic terminal alkynes with varied lengths of carbon chain gave corresponding trialkyl substituted alkenes in good yields (**4a**-**4c**).

Aliphatic alkynes tethered with chloride, ester, amides are tolerated in this method, furnishing corresponding functionalized alkenes in synthetic useful yields, rendering further elaboration opportunities for the products (**4d**-**4h**). The reaction could be applied to late-stage functionalization of natural products and

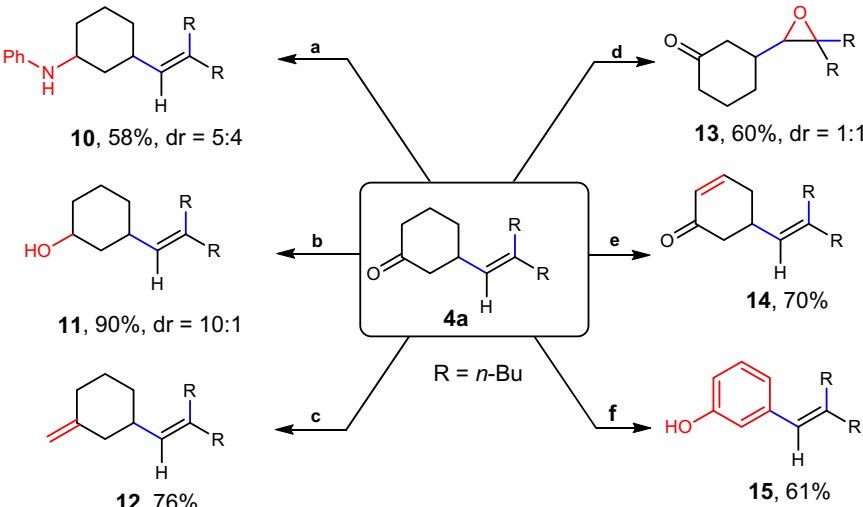

**Fig. 4 Synthesis of unfunctionalized tri- and tetrasubstituted alkenes.** These results indicate that product of dialkylation of alkynes can be sufficiently transformed into unfunctionalized alkenes. Yields are for isolated and purified products. The detailed procedures have been presented in Supplementary Information (SI).

**Fig. 5 Synthetic applications of 4a.** Reaction conditions: **a** Reductive amination. (1) PhNH₂, MgSO₄, AcOH, DCM (0.2 M), rt, 2 h; (2) NaBH₃CN, MeOH, rt, 10 h. **b** Reduction. NaBH₄, MeOH, rt, 2 h. **c** Wittig Olefination. n-BuLi, Ph₃PCH₂Br, THF, −30 °C to rt, overnight. **d** Epoxidation. m-CPBA, NaHCO₃, DCM, 0 °C, 2 h. **e** Desaturation. Pd(TFA)₂, 1 atm O₂, DMSO, 85 °C, 14 h. **f** Aromatization. Pd(TFA)₂, PTSA, N,N-dimethylpyridin-2-amine, 1 atm O₂, DMSO, 80 °C, 24 h.

drug molecules. Estrone and oxaprozin derived terminal alkynes could be transformed to corresponding trialkylated products in 52% and 57% yields, respectively (**4i** and **4j**). Notably, the reaction conditions could be applied to gram-scale synthesis (10.0 mmol scale) without erasing the efficiency of the reaction, affording 1.68 g of **4a** in 67% yield. Second, terminal aromatic alkynes were tested. Electron-donating and electron-withdrawing substituted aryl alkynes are all good substrates for this transformation with **A2** (20 mol%) and zinc bromide (10 mol%) as Lewis acid. In these cases, 2-methyl-2-butanol was used as proton source instead of water. Terminal aryl alkynes with *para-*, *meta-*, and *ortho-*substitution patterns could be converted to corresponding alkenes with two aliphatic substituents in moderate to good yields (**5a**-**5k**). It is noteworthy that alkynyl arylhalides are compatible in this reaction with halides intact (**5f** and **5g**), leaving a chemical handle to build molecular complexity by cross-coupling reactions. The structure and stereochemistry of the product were determined unambiguously by X-ray diffraction analysis of **5c**. Alkynyl indole could be converted to substituted vinyl indole in synthetic useful yield (**5l**). Third, internal alkynes were also examined, which could lead to synthetically challenging tetrasubstituted alkenes. Internal alkynes are challenging

substrates for Ni-catalyzed carbo-difunctionalization of alkynes due to their steric hindrance. It is noteworthy that internal alkynes are compatible in this reaction. Diverse substituted 1-phenyl-2-alkylalkynes are good substrates under the reaction conditions, giving trialkylaryl-substituted alkenes (**6a**-**6e**) in good yields with excellent regioselectivities. The regio- and stereo-chemistry was confirmed by X-ray diffraction analysis of **6b**. Stereodefined alkenes with four aliphatic substitutents could be obtained using this protocol in good yields with exclusive regio- and stereo-selectivity (**6f**-**6i**).

Next, we evaluated the scope of two electrophiles (Fig. 3). A wide variety of alkyl bromides proved amenable to this reaction, substantially expanding the synthetic potential of this protocol (Fig. 3, top). Alkyl bromides with different lengths of aliphatic chain as well as alkyl or aryl ethers, chlorides, could be combined with terminal and internal alkynes, affording tri- and tetrasub-stituted alkenes in moderate to good yields (**7a**-**7l**). Notably, the reaction demonstrated excellent selectivity over alkynes and alkenes. Alkene or alkyne tethered alkyl bromides underwent cross-electrophile three-component reaction smoothly, delivering corresponding multiple substituted alkenes in 68% and 65% yields, respectively, leaving the double bond and triple bond

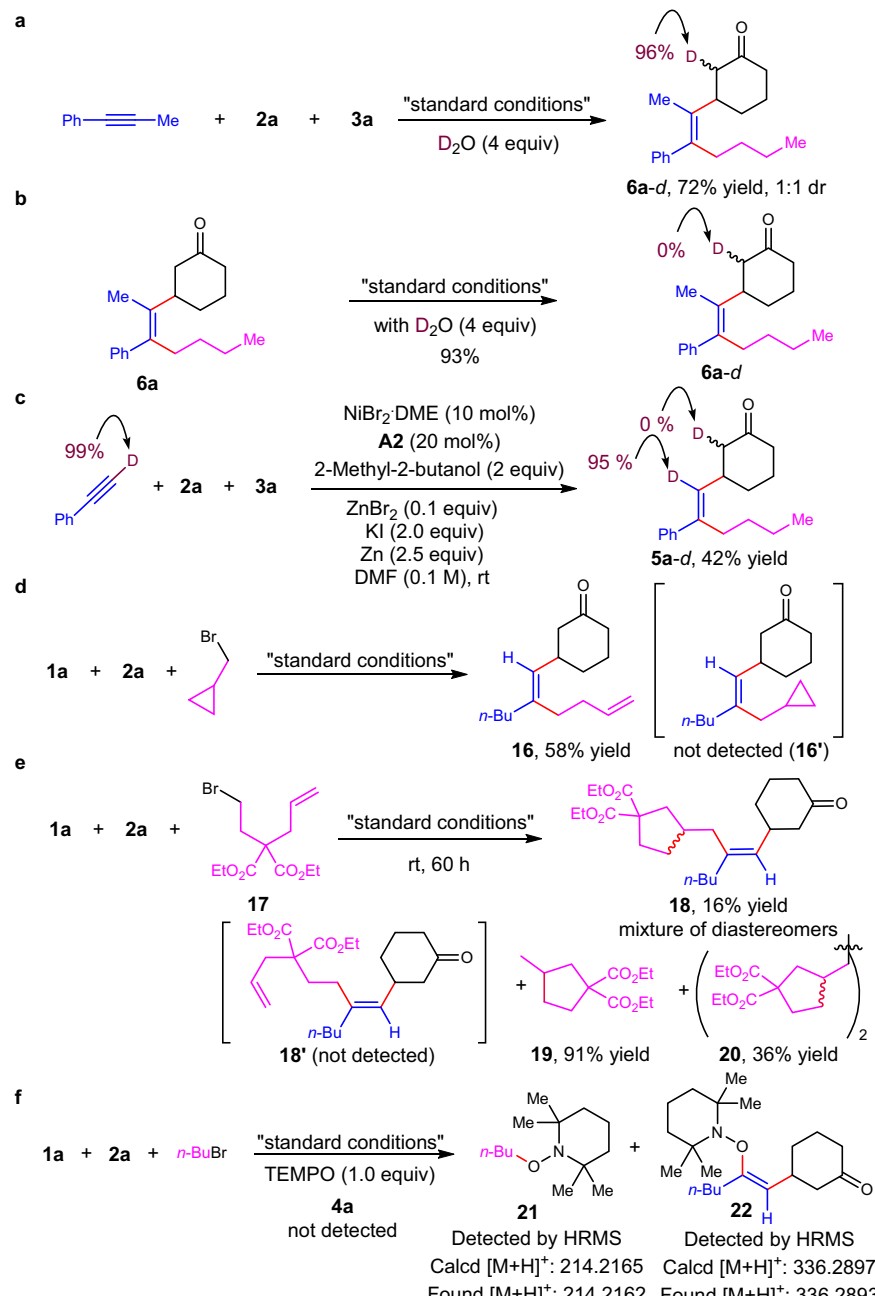

**Fig. 6 Mechanistic investigations and control experiments. a** Reaction in the presence of deuterated water. **b** Treat the product **6a** under reaction conditions with deuterated water. **c** The reaction of deuterated phenylacetylene. **d** Radical clock reaction using cylcopropylmethylbromide. **e** Radical clock reaction using **17**. **f** Radical trap with TEMPO.

intact (**7m** and **7n**). Moreover, functional groups, such as ester and amide, which are sensitive to organometallic reagents, could be tolerated under the reaction conditions (**7o-7r**).

Secondary alkyl halides, such as 4-iodotetrahydro-2*H*-pyran, could be converted to corresponding alkene **7s** in 52% yield. Unfortunately, tertiary alkyl halides could not form the desired product. Additionally, various cyclic and acyclic enones could be involved in this cross-electrophilile dialkylation process (Fig. 3, bottom). Cycloheptenone could be coupled with alkyne and *n*-butylbromide to give **8a** in 61% yield. A couple of acyclic enones were also good substrates in this three-component reaction, furnishing corresponding multialkylated alkenes with regio- and stereocontrol in 53%-61% yields (**8b-8e**). Cyclohexenones with diverse bulky substitutions were also tolerated in the reaction,

delivering the desired products (**8f** and **8g**) in 52% and 66% yields, respectively. Acyclic enones with more steric hindered substituents and acryamides could be involved in the reaction, albeit in lower effeciency (**8h** and **8i**). The major competing side reactions include the alkylalkynylation (**8h′**)and hydroalkylation of alkynes (**8i′**) (Supplementary Figs. 11–18).

**Derivatization.** Unfunctionalized alkenes with multiple sub-stitutions are important yet difficult to access[13]. The resulting alkenes could be transformed into unfunctionalized alkenes by removing the carbonyl group in good yields (**9a-9f**), providing a useful method to tri- and tetrasubstituted all-carbon alkenes without any functional group (Fig. 4). Notably, this protocol

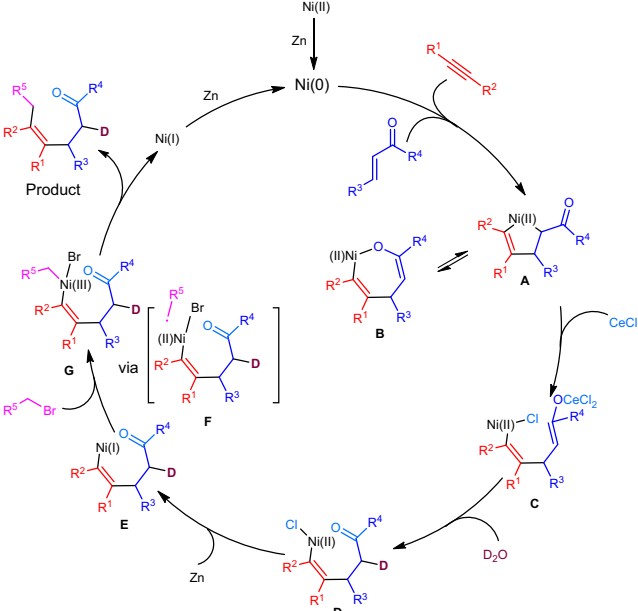

**Fig. 7 Proposed reaction mechanism.** Specifically, **A**, **B**: Ni (II) intermediates of cyclometallation of alkynes with enones; **C**, **D**: Ni (II) intermediates after Ni-O bond dissociation; **E**: Ni (I) intermediate reduced by Zn; **F**: transition state for single-electron reduction of Ni (I) with alkyl bromides; **G**: Ni (III) intermediate of oxidative addition to alkyl bromides.

provides a convenient access to unsymmetrical all-carbon sub-stituted alkenes with up to four aliphatic substituents in a regio- and stereocontrolled fashion, which are inaccessible otherwise.

To further demonstrate the utility of this methodology, an array of synthetic applications of the products were performed based on **4a** (Fig. 5). By reductive amination or reduction of the carbonyl group, **4a** could be converted to homoallyic amine (**10**) and alcohol (**11**) in 58% and 90% yields, respectively. Skipped diene **12** was obtained in 76% yield by Wittig reaction. Epoxidation of the double bond furnished **13** in 60% yield using *m*-CPBA as the oxidant. The product could undergo site-selective desaturation to give unconjugated dienone **14** in 70% yield. Treating **4a** under oxidative conditions underwent aromatization to deliver vinyl free phenol **15** in 61% yield.

**Mechanistic consideration**. To shed light on the detail of this sequential three-componet reaction, we set up a series of experi-ments to probe the reaction mechanism (Fig. 6). First, 1-phenyl-1-propyne was exposed to the reaction conditions in the presence of $D_2O$ (4 equiv) in lieu of $H_2O$, **6a**-*d* was obtained in 72% yield with 96% deuteration of single proton at $\alpha$-postion to carbonyl as a mixture of diastereomers. While no deuterium incorpoartion was observed when **6a** was subjected to the reaction conditions in the presence of deuterated water. When deuterated phenylacetylene was subjected to the reaction conditions, **5a**-*d* was obtained in 42% yield. No deuterium scrambling from terminal position of pheny-lacetylene to $\alpha$-postion to carbonyl was observed. These results indicate one $\alpha$-proton of ketone is from water during the reaction course and non-exchangable with surroundings. Next, radical clock experiments were carried out. When (bromomethyl)cyclopropane was used in the reaction, skipped dienone **16** was obtained in 58% yield by ring-opening/cross-coupling reaction process without the formation of **16'**. Moreover, homoallylic bromide **17** could be transformed into **18** via radical cyclization/cross-coupling process in 16% yield along with the formation of byproducts **19** and **20**[57]. The acyclic product **18'** was not detected. In addition, the use of TEMPO completely shut down the desired reaction. The formation

of **21** and **22** were detected by mass. No *n*-butyl zinc species was detected during the reaction course (Supplementary Figs. 5–9). The combined results suggest the involvement of alkyl radical species in this reaction.

**Proposed reaction mechanism**. Based on the experimental results and literature, a proposed mechanism of this reaction is depicted in Fig. 7. First, Ni (0) species generated from Ni (II) by reduction initiated metallacyclic intermediate **A** via cross-oxidative cyclometallation between alkyne and enone. C-bound intermediate **A** could equilibrate with O-bound intermediate **B**. In the presence of Lewis acid ($CeCl_3$ or $ZnBr_2$), intermediate **C** was formed via transmetalation, which could further undergo protonation to give vinylnickel intermediate **D** with proton source. The presence of Lewis acid might faciliate the generation of **C** by forming M-O bond (M = Ce or Zn). **D** could be reduced by zinc to give Ni (I) intermediate **E**, which could undergo oxi-dative addition to alkyl bromide to form Ni (III) intermediate **G** via single stepwise electron transfer and radical recombination with **F**. Intermediate **G** underwent reductive elimination to form the final product along with Ni(I) intermediate, which could be further reduced by Zn to complete the catalytic cycle.

## Discussion

In summary, we have demonstrated a Ni-catalyzed fully inter-molecuar regio- and stereo-selective cross-dialkylation of alkynes in the presence of alkyl halides and $\alpha,\beta$-unsaturated compounds, affording stereodefined tri- and tetrasubstituted alkenes with multiple alkyl substitutions[58–62]. Notably, the reaction enables *syn*-selective dialkylation of alkynes with exclusive regioselectivity in the absence of any direction group, providing an alternative to access multi-substituted unfunctionalized alkenes from readily-accessible reagents. The use of reductive conditions allows for the reaction proceed at room temperature, circumventing the use of air- and moisture-sensitive organometallic reagents.

## Methods

**General procedure for the fully intermolecular dialkylation of alkynes.** The reaction was operated in a nitrogen-filled glove box. 4-Cyanopyridine (5 mol%), zinc (4.5 equiv), $CeCl_3$ (0.8 equiv), KI (2.0 equiv), $NiBr_2 \cdot DME$ (10 mol%), alkyne (2.0 equiv, if solid), and alkyl bromide (2.0 equiv, if solid) were added into an oven-dried vial containing a magnetic stirring bar. If the alkyne or alkyl bromide is a solid, it was added along with other solid reagents. Anhydrous DMF (0.1 M) was added and rapid stirring was commenced. Then $H_2O$ (1.5 equiv), alkyne (2.0 equiv, if liquid), enone (0.2 mmol), and alkyl bromide (2.0 equiv, if liquid) were added sequentially via syringe. The vial was sealed with parafilm and stirred vigorously for 48 h. The reaction was diluted with ethyl acetate (30 mL) and washed with brine (50 mL). The aqueous layer was extracted twice with ethyl acetate (20 mL). The combined organic layer was dried over magnesium sulfate, filtered, evaporated and purified by silica gel chromatography with hexanes/ethyl acetate as eluent to give the corresponding product in pure form.

## Data availability

The authors declare that all other data supporting the findings of this study are available within the article and Supplementary Information files, and also are available from the corresponding author upon reasonable request. The X-ray crystallographic coordinates for structures reported in this study have been deposited at the Cambridge Crystallographic Data Centre (CCDC), under deposition numbers of CCDC 1983359 and CCDC 1983360. These data can be obtained free of charge from The Cambridge Crystallographic Data Centre via www.ccdc.cam.ac.uk/data_request/cif.

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

## Acknowledgements
We sincerely acknowledge NSFC (21971101 and 21801126), Guangdong Basic and Applied Basic Research Foundation (2019A1515011976), Thousand Talents Program for Young Scholars, The Pearl River Talent Recruitment Program (2019QN01Y261), and Guangdong Provincial Key Laboratory of Catalysis (No. 2020B121201002) for financial support. We thank Dr. Xiaoyong Chang (SUSTech) for X-ray crystallographic analysis. We acknowledge the assistance of SUSTech Core Research Facilities, and Shan Wang (SUSTech) for reproducing the results of **4e**, **5h**, **6h**, and **7j**.

## Author contributions
Y.Z.Z. discovered and developed the reaction. W.S. conceived and designed the project. Y.Z.Z. and N.X. performed the experiments and collected the data. W.S. wrote the manuscript with contributions from all authors.

## Competing interests
The authors declare no competing interests.
