## [Peer Review File · Nature Communications]

REVIEWER COMMENTS

Reviewer #1 (Remarks to the Author):

Shu and coworkers report a three-component approach to the dialkylation of alkynes through Ni-catalyzed cross-electrophile couplings between alkynes, enones, and alkyl bromides. Utilizing a similar cyclization strategy developed by Montgomery (J. Am. Chem. Soc. 2018, 140, 7074), this transformation proceeded via a Ni(0)-mediated oxidative cyclometallation between alkyne and enone, followed by cross-coupling with alkyl bromide. A series of multiple substituted alkenes were generated with regio- and stereoselectivity under mild conditions. Furthermore, several synthetic applications based on ketones and alkenes have been demonstrated, and a series of mechanistic experiments have been conducted to disclose the potential reaction pathway. However, the current reaction efficiency is somewhat too low for a useful and general methodology to be published in Nat. Commun. The following concerns should be addressed thoroughly before any resubmission.

1) The current efficiency of this three-component reaction is too low, and most of the isolated yields are around 40%-60%, even with excessive amounts of alkynes and alkyl bromides. For a new and useful methodology to be published in Nat. Commun., such low efficiency is quite frustrating. Critically, the author claimed that exclusive regio- and stereo-selectivity was obtained for this transformation, while the low yields made this claim dubious. What are the side products? The authors should comment on mass balance and identify any side products.

2) Only simple enones have been explored, how about the reactivity of enals and other electron-deficient alkenes? And what happened if secondary and tertiary alkyl precursors were employed? Is there alkyl zinc intermediate involved?

3) The isolated yield of 18 is only 16%, pretty low compared to yields of 7m and 7o (68%, 50%), which seems unreasonable. Are there any other side products? What's the rate for the ring-close step? It is necessary to figure out the major byproducts in this case.

4) The authors used 10 mol% of Ni(II) and 5 mol% of 4-cyanopyridine or 2.5 mol% L3 as a ligand. In the proposed mechanism part, they claimed that "ligand supported Ni(0) species ...". Did the solvent act as ligand? What the real role of 4-cyanopyridine is, specifically in the presence of Lewis acid additives such as CeCl₃? Additives such as ZnBr₂ and KI were also used. Did the addition of these additives inhibit any side products?

5) In some cases, 2-methyl-2-butanol instead of H₂O was employed. The authors should comment on it.

6) several minor mistakes: Page 6, "region- and stereoselectivity"

Page 7, last line, "with upto"

Page 9, line 6, "To Moreover, "

Page 9, "C-Bound"

Reviewer #2 (Remarks to the Author):

Alkynes are one of the most common and important structural motifs. Accordingly, the difunctionalization of alkynes plays an important role in accessing multiple substituted alkenes. The manuscript by Shu and coworkers describes an interesting intermolecular cross-dialkylation of terminal and internal alkynes, alkyl halides, and enones, providing an excellent way to prepare tri- and tetrasubstituted alkenes with up to four aliphatic substituents in regio- and syn-selective manner at room temperature. The ketone group tethered on the resultant products enables further

diverse functionalization and increased complexity. The avoidance of sensitive organometallic reagents and the mild reductive conditions allows for a broad substrate scope with good functional group tolerance (-Br, -Cl, double bond, second internal triple bond were all well tolerated, natural products and drug molecules 4i and 4j were also applied). Also, the achievement of stereodefined alkenes with four aliphatic substituents is notable. A series of further transformations of the generated ketone containing alkenes demonstrate the utility of this three-component cascade protocol. Deuteration and radical clock experiments explain the source of the proton and the involvement of the alkyl radical, which is in line with the proposed mechanism.

Overall, I recommend publication of this highly interesting work in *Nat. Commun.* as it describes leading research in a topical area which will be of interest for a broad readership.

The following minor points have to be addressed:

1. In page 2, "...the second Csp³-Csp³ bond." should be "...the second Csp²-Csp³ bond."
2. Next to the diarylation (ref. 19-29) arylsulfonylation (*Nat. Catal.*, 2019, 2, 678-687) has been reported. The authors may want to cite that manuscript. There is also a review on that topic *Chem. Sci.*, 2020, 11, 4051.
3. The reaction condition for the scope of terminal aryl alkynes in Fig. 2 is different from the standard condition, it's better to mention it in the text.
4. What is the role of 2-methyl-2-butanol as an additive in the Fig. 2b and 3a (scope) ?
5. The color of the methyl group in scope 7d should be changed to purple.
6. The "Bu-n" substituent in Fig. 5, compound 11 should be corrected.
7. The Ni-Br bonds should be added in intermediates H and G in Figure 7.

June 30th, 2020
Magnus Rueping

Reviewer #3 (Remarks to the Author):

Zhan and coworkers reported the methodology of intermolecular cross-dialkylation of alkynes catalyzed by nickel in a syn-selective fashion without any directing or activating groups. The main feature of this protocol is the synthesis of multifunctionalized alkene in a regio- and stereoselective manner. The importance of this kind of reaction lies in further modification for the synthesis of natural products, bioactive molecules and material sciences. In addition, substituted alkenes can also serve as useful precursors for other complex functional groups and chiral centres. This reaction is also applicable to a broad substrate scope which can give the moderate to high yield of desired product. In addition, they proposed the general mechanistic pathways for the reaction and provided supporting experiments.

From the point of view of reaction itself, the reaction condition is relatively mild in terms of room temperature condition and the employment of Ni catalyst without any other organometallic reagents. As a result, both regio and stereoselectivity can be achieved in the final product at this mild condition.

Some points should be corrected or clarified further before being accepted:

1. It involves free-radical reaction. But the authors only performed one control experiment. Could they please try other control experiments with TEMPO and BHT.
2. Combine ligand and catalyst together to pre-synthesize the active species and see if it's more efficient.
3. Since the active species is Ni(0) in the system, what if just using Ni(0) as pure catalyst.
4. In addition, some recent key references seem to be missing. Please cite (*Chem. Sci.*, 2019, 10, 10417), (*ACS Catal.* 2019, 9, 9199) and (*Angew. Chem. Int. Ed.* 2020, 59, 1).

Overall, It's suitable for nature communication with the above revisions.

Point-to-Point Response to Reviewers' Comments

Reviewer: 1

Comments:

Shu and coworkers report a three-component approach to the dialkylation of alkynes through Ni-catalyzed cross-electrophile couplings between alkynes, enones, and alkyl bromides. Utilizing a similar cyclization strategy developed by Montgomery (J. Am. Chem. Soc. 2018, 140, 7074), this transformation proceeded via a Ni(0)-mediated oxidative cyclometallation between alkyne and enone, followed by cross-coupling with alkyl bromide. A series of multiple substituted alkenes were generated with regio- and stereoselectivity under mild conditions. Furthermore, several synthetic applications based on ketones and alkenes have been demonstrated, and a series of mechanistic experiments have been conducted to disclose the potential reaction pathway. However, the current reaction efficiency is somewhat too low for a useful and general methodology to be published in Nat. Commun. The following concerns should be addressed thoroughly before any resubmission.

General response: We thank the reviewer for insightful comments! We have improved the efficiency of some substrates by further optimization the reaction conditions. Most of the substrates suffered from moderate yields were improved by 10-20% yield. The concerns of this reviewer have been addressed by additional experiments and rationalization in the revised manuscript. Please see the details as following:

Comment 1: The current efficiency of this three-component reaction is too low, and most of the isolated yields are around 40%-60%, even with excessive amounts of alkynes and alkyl bromides. For a new and useful methodology to published in Nat. Commun., such low efficiency is quite frustrating. Critically, the author claimed that exclusive regio- and stereo-selectivity was obtained for this transformation, while the low yields made this claim dubious. What are the side products? The authors should comment on mass balance and identify any side products.

Our response: This fully intermolecular reductive dialkylation of alkynes is more challenging compared to intramolecular version of reactions as demonstrated in Figure 1c. The intermolecular three-component reaction is also more practical due to the ready availability of starting materials. The major identified side products include the oligomerization of enones, Michael addition of alkyl halides onto enones, alkylalkynylation of alkyne. We have identified a couple of major competitive by-

products in the reaction. Please see section 4.6 in the Supporting Information. ¹H NMR spectroscopy and mass spectroscopy of crude mixture were used to determine the regio- and stereoselectivity of the reaction. No detectable isomers were observed. Furthermore, great effort has been paid to improve the efficiency of the reaction. Yield of most of the substrates suffering from low-yielding was further improved by 10-20% after further optimization of the reaction conditions (**5b**, **5c**, **5f**, **5h**, **5i**, **5j**, **5l**, **7b**, **7c**, **7f**, **7h**, **7i**, **7n**, **7o**, **8a**, **8b**, **8c**, **8d**, **8e**).

Comment 2: Only simple enones have been explored, how about the reactivity of enals and other electron-deficient alkenes? And what happened if secondary and tertiary alkyl precursors were employed? Is there alkyl zinc intermediate involved?

Our response: First of all, more electron-deficient alkenes have been evaluated, and **8f-8i** are obtained in synthetic useful yields. Unfortunately, enals are not compatible in the reaction due to the fast side reactions of enals under the reaction conditions. Second, secondary and tertiary alkyl precursors were tested. Secondary alkyl precursors could be successfully employed in this fully intermolecular cross-electrophile dialkylation process, delivering the desired product **7s** in 52% yield. Unfortunately, tertiary alkyl precursors proved to be unsuccessful. Accordingly, a comment “Secondary alkyl halides, such as 4-iodotetrahydro-2*H*-pyran, could be converted to corresponding alkene **7s** in 52% yield. Unfortunately, tertiary alkyl halides could not form the desired product.” is added to the revised manuscript. Third, a series of experiments were set up to probe whether alkyl zinc intermediate was involved in the reaction (Section 4.4, Supporting Information). No butyl zinc species was detected using 1-bromobutane under standard reaction conditions with or without water. In addition, the reaction was in situ monitored by ¹H NMR using 1-bromobutane, no signal of *n*-butyl zinc species was found during the reaction process. To this end, alkyl zinc species is not likely to be involved in the reaction.

Comment 3: The isolated yield of **18** is only 16%, pretty low compared to yields of **7m** and **7o** (68%, 50%), which seems unreasonable. Are there any other side products? What’s the rate for the ring-close step? It is necessary to figure out the major byproducts in this case.

Our response: The low yielding of **18** attributes to the steric effect of α -branched radical intermediate from radical ring-closing, which is more steric hindered and results in a sluggish and less efficient rebound to metal-center. Following the suggestion of the reviewer, we have identified the major by-products of **17**. Besides the three-component

cross-coupling product **18**, major byproducts including cyclization products **19** and **20** were detected (Figure 6e). According to literature, the radical ring-closing step is very fast ($>5 \times 10^6 \text{ s}^{-1}$, ref. *Aust. J. Chem.* **1983**, *36*, 545-556).

Comment 4: The authors used 10 mol% of Ni(II) and 5 mol% of 4-cyanopyridine or 2.5 mol% L3 as a ligand. In the proposed mechanism part, they claimed that “ligand supported Ni(0) species ...”. Did the solvent act as ligand? What the real role of 4-cyanopyridine is, specifically in the presence of Lewis acid additives such as CeCl₃? Additives such as ZnBr₂ and KI were also used. Did the addition of these additives inhibit any side products?

Our response: We systematically evaluated the effect on the loading of 4-cyanopyridine, indicating that 5 mol% of 4-cyanopyridine afforded the optimal yield (Table S1). Thus, the role of 4-cyanopyridine is adjusted to be an additive, which might facilitate the cyclometallation step of two unsaturated bonds to form intermediate **A** according to literature (*J. Organometallic Chem.* **1989**, *375*, 259-264). Accordingly, we have removed the “ligand supported” from the text. The use of additives did improve the yield of the desired intermolecular three-component cross-dialkylation product of alkynes (Table 1, entries 2, 10 and 12). We rationalize the role of CeCl₃ and ZnBr₂ to be similar, which facilitate the dissociation of Ni-O/Ni-C bond of intermediate **A/B** to form intermediate **C**. A comment “The presence of Lewis acid might facilitate the generation of **C** by forming M-O bond (M = Ce or Zn).” is added to the manuscript. The role of KI is to undergo halide exchange to generate alkyl iodide in situ slowly from alkyl bromide to maintain alkyl iodide in low concentration during the reaction course. A comment is added to the manuscript. As shown in the Table 1, additive CeCl₃ and KI significantly improve the yield of desired three component cross-coupling dialkylation product.

Comment 5: In some cases, 2-methyl-2-butanol instead of H₂O was employed. The authors should comment on it.

Our response: As suggested, a comment “In these cases, 2-methyl-2-butanol was used as proton source instead of water.” is added to the revised manuscript.

Comment 6: several minor mistakes: Page 6, “region- and stereoselectivity” Page 7, last line, “with upto” Page 9, line 6, “To Moreover,” Page 9, “C-Bound”

Our response: We have corrected the errors as suggested in the revised manuscript. We thank the reviewer for insightful comments and suggestions to improve the

manuscript. With the extra efforts for corrections and revisions, we hope the reviewer find the revised manuscript suitable for publication in Nature Communications.

Reviewer: 2

Comments:

Alkynes are one of the most common and important structural motifs. Accordingly, the difunctionalization of alkynes plays an important role in accessing multiple substituted alkenes. The manuscript by Shu and coworkers describes an interesting intermolecular cross-dialkylation of terminal and internal alkynes, alkyl halides, and enones, providing an excellent way to prepare tri- and tetrasubstituted alkenes with up to four aliphatic substituents in regio- and syn-selective manner at room temperature. The ketone group tethered on the resultant products enables further diverse functionalization and increased complexity. The avoidance of sensitive organometallic reagents and the mild reductive conditions allows for a broad substrate scope with good functional group tolerance (-Br, -Cl, double bond, second internal triple bond were all well tolerated, natural products and drug molecules 4i and 4j were also applied). Also, the achievement of stereodefined alkenes with four aliphatic substituents is notable. A series of further transformations of the generated ketone containing alkenes demonstrate the utility of this three-component cascade protocol. Deuteration and radical clock experiments explain the source of the proton and the involvement of the alkyl radical, which is in line with the proposed mechanism.

Overall, I recommend publication of this highly interesting work in Nat. Commun. as it describes leading research in a topical area which will be of interest for a broad readership.

General response: We thank the reviewer for supporting the publication of this work in Nature Communications. The concerns of this reviewer are addresses as below.

Comment 1: In page 2, "...the second Csp3-Csp3 bond." should be "...the second Csp2-Csp3 bond."

Our response: We have corrected the error as suggested.

Comment 2: Next to the diarylation (ref. 19-29) arylsulfonylation (Nat. Catal., 2019, 2, 678-687) has been reported. The authors may want to cite that manuscript. There is also a review on that topic Chem. Sci., 2020,11,4051.

Our response: The mentioned references are cited as refs. 30 and 31 in the revised manuscript.

Comment 3: The reaction condition for the scope of terminal aryl alkynes in Fig. 2 is different from the standard condition, it's better to mention it in the text.

Our response: A comment "Electron-donating and electron-withdrawing substituted aryl alkynes are all good substrates for this transformation with **A2** (20 mol%) and zinc bromide (10 mol%) as lewis acid. In these cases, 2-methyl-2-butanol was used as proton source instead of water" is added to the revised manuscript. Thanks very much for the suggestion!

Comment 4: What is the role of 2-methyl-2-butanol as an additive in the Fig. 2b and 3a (scope) ?

Our response: 2-Methyl-2-butanol in Figs. 2b and 3a plays similar role of water as shown in Figure 7 to facilitate the dissociation of M-O bond of intermediate **C** to give intermediate **D**. When CeCl₃ was used, the use of water gave better result. When ZnBr₂ was used, the use of 2-methyl-2-butanol gave better yield. A comment "In these cases, 2-methyl-2-butanol was used as proton source instead of water" is added to the text".

Comment 5: The color of the methyl group in scope 7d should be changed to purple.

Our response: We have changed the color of methyl group in 7d to purple as suggested.

Comment 6: The "Bu-n" substituent in Fig. 5, compound 11 should be corrected.

Our response: We have corrected the "Bu-n" substituent to "*n*-Bu" in Figure 5 and compound **11** was corrected as suggested.

Comment 7: The Ni-Br bonds should be added in intermediates H and G in Figure 7.

Our response: Ni-Br bonds have been added to intermediates H and G in Figure 7 as suggested.

Reviewer: 3

Comments: Zhan and coworkers reported the methodology of intermolecular cross-dialkylation of alkynes catalyzed by nickel in a syn-selective fashion without any directing or activating groups. The main feature of this protocol is the synthesis of multifunctionalized alkene in a regio- and stereoselective manner. The importance of this kind of reaction lies in further modification for the synthesis of natural products, bioactive molecules and material sciences. In addition, substituted alkenes can also

serve as useful precursors for other complex functional groups and chiral centres. This reaction is also applicable to a broad substrate scope which can give the moderate to high yield of desired product. In addition, they proposed the general mechanistic pathways for the reaction and provided supporting experiments.

From the point of view of reaction itself, the reaction condition is relatively mild in terms of room temperature condition and the employment of Ni catalyst without any other organometallic reagents. As a result, both regio and stereoselectivity can be achieved in the final product at this mild condition.

Overall, It's suitable for nature communication with the above revisions.

General response: We thank the reviewer for supporting the publication of this work in Nature Communications. The concerns of this reviewer are addresses as below.

Comment 1: Some points should be corrected or clarified further before be accepted:

1. It involves free-radical reaction. But the authors only performed one control experiment. Could they please try other control experiments with TEMPO and BHT.

Our response: We thank the reviewer for the kind suggestions! Control experiments with TEMPO and BHT are tested. The use of TEMPO completely shut down the desired reaction, with intermediates trapped by TEMPO (Fig. 6f). The use of BHT slightly diminished the efficiency of the reaction, giving **4a** in 57% yield (Section 4.3, Supporting Information). In this case, BHT probably serves as proton source.

Comment 2: Combine ligand and catalyst together to pre-synthesize the active species and see if it's more efficient.

Our response: The pre-synthesized catalyst by combining ligand and catalyst was tried in the reaction. It gave desired product **4a** in 63% yield, which is similar to the catalytic system formed in situ. A comment 'Similar result was obtained when presynthesized precatalyst was used, giving the desired product **4a** in 63% yield.' is added to the revised manuscript.

Comment 3: Since the active species is Ni(0) in the system, what if just using Ni(0) as pure catalyst.

Our response: The use of Ni(COD) as Ni (0) species was conducted under otherwise identical to standard conditions, giving **4a** in 36% yield. Please see Table 1, entry 8 in the revised manuscript.

Comment 4: In addition, some recent key references seem to be missing. Please cite (Chem. Sci., 2019, 10, 10417), (ACS Catal. 2019, 9, 9199) and (Angew. Chem. Int. Ed. 2020, 59,1).

Our response: We have added the mentioned references (Chem. Sci., 2019, 10, 10417; ACS Catal. 2019, 9, 9199) as ref. 60 and 61. This reference (Angew. Chem. Int. Ed. 2020, 59, 6466) is not found according to the information.

We thank all the reviewers for insightful comments and suggestions!

REVIEWERS' COMMENTS

Reviewer #1 (Remarks to the Author):

All requested revisions from reviewers have been properly made. Therefore, this referee would like to recommend this revised publication in Nature Communications.

Reviewer #2 (Remarks to the Author):

The authors provided a revised manuscript in which they addressed the comments of the reviewers. After carefully reading the comments of the reviewers, the response of the author and the revised manuscript I recommend acceptance of this highly interesting work in Nat. Commun. as it describes leading research in a topical area which will be of interest for a broad readership.

All comments have been addressed, the authors have carried out additional experiments, added clarifications and thus improved the manuscript. Publication of this fine work is recommended.